# Fishery Anesthetics in Aquaculture Products: Safety Concerns and Analytical Methods

**DOI:** 10.3390/foods14223928

**Published:** 2025-11-17

**Authors:** Bao-Zhu Jia, Xue-Ying Rui, Yu Wang, Xi Zeng, Shu-Jing Sheng, Bi-Jian Zeng, Zhen-Lin Xu, Lin Luo

**Affiliations:** 1College of Biology and Food Engineering, Guangdong University of Education, Guangzhou 510303, China; jbzjbz130@163.com (B.-Z.J.); shengshujing2008@163.com (S.-J.S.); bijianzeng@163.com (B.-J.Z.); 2Guangdong Provincial Key Laboratory of Food Quality and Safety, South China Agricultural University, Guangzhou 510642, China; micadeemail@163.com (X.-Y.R.); jallent@163.com (Z.-L.X.); 3Guangzhou Institute of Food Inspection, Guangzhou 511400, China; xxwangyu@163.com (Y.W.); lovelyzeng@126.com (X.Z.)

**Keywords:** aquatic products, fishery anesthetics, analytical methods, food safety

## Abstract

Fishery anesthetics are extensively employed in aquaculture to mitigate stress and reduce mortality during handling, transportation, and processing of farmed fish. While they enhance operational efficiency and reduce economic losses for fish merchants, the potential residual presence of these anesthetics raises concerns regarding human health risks and environmental impact. This review examines six commonly used anesthetics in aquaculture—eugenol, MS-222, benzocaine, 2-phenoxyethanol, diazepam, and quinaldine—focusing on their mechanisms of action, application risks, ecotoxicological effects, and methods for residue analysis. The objective is to promote the safe and informed application of these anesthetics, mitigate their biological toxicity, and minimize their ecological impact. Furthermore, the review provides technical insights into monitoring and managing anesthetic residues in aquaculture to ensure the safety of aquatic products and safeguard environmental health while also supporting the development of sustainable aquaculture practices.

## 1. Introduction

Fresh aquatic products are an important source of high-quality protein and essential nutrients, playing a valuable role in a balanced human diet [1]. Owing to their desirable flavor, high nutritional value, and low cholesterol content, these products are widely consumed worldwide. However, with the expansion of the seafood market, ensuring the safety of aquatic products has become a major concern. Consumers are increasingly attentive not only to nutritional benefits but also to potential hazards such as chemical residues, contaminants, and improper handling practices.

The application of fishery anesthetics is a critical practice to mitigate stress during aquaculture operations [2]. Procedures such as transport and handling can induce intense stress, leading to injury, mortality, and compromised product quality [3]. The judicious use of anesthetics helps mitigate these adverse effects, supports animal welfare, and enhances antioxidant capacity, thereby extending shelf life [4,5]. However, it is essential to note that the safety and efficacy of these anesthetics are dose-dependent. Understanding their physiological impacts and establishing optimal dosage regimes are crucial to safeguarding both fish health and consumer safety.

Despite the clear benefits, the use of anesthetics in many regions remains insufficiently regulated. Dosages and withdrawal periods are often determined empirically, which increases the risk of unsafe residue levels in edible tissues. Moreover, regulatory standards for maximum residue limits (MRLs) and withdrawal periods vary significantly across countries, and some anesthetics still lack established guidelines (Table 1). These regulatory gaps highlight the urgent need for more research on rapid detection methods, residue elimination kinetics, and anesthetic efficacy.

This review systematically identifies and analyzes relevant studies on fishery anesthetics, focusing on their safety, residue detection, and implications for food safety. A comprehensive literature search was conducted using keywords such as “fishery anesthetics,” “aquaculture anesthetics,” “residue detection,” and “maximum residue limits,” combined with Boolean operators, across databases including PubMed, Scopus, Web of Science, Google Scholar, and ScienceDirect. Studies published in peer-reviewed journals over the past three decades were included, as long as they addressed the use, regulation, safety profiles, and residue detection methods of anesthetics, incorporating both developed and developing countries. Non-peer-reviewed literature and irrelevant studies were excluded, and duplicate entries were removed during full-text screening. This methodology ensured the inclusion of relevant and up-to-date studies, providing a thorough analysis of commonly used anesthetics, recent advances in residue depletion, and detection techniques. Furthermore, the review identifies research gaps and suggests future directions to strengthen regulatory frameworks and safeguard public health.

## 2. Overview of Fishery Anesthetics

Fishery anesthetics are chemical agents employed in aquaculture to induce sedation or temporary immobility in farmed fish, thereby reducing stress and physical injury during operations such as transport, handling, sampling, and artificial reproduction [12,13]. By suppressing stress responses, these compounds help maintain fish welfare, lower mortality, decrease disease susceptibility, and preserve flesh quality [14]. Stress-related metabolic changes—such as lactic acid accumulation—can impair meat texture and overall product quality; anesthetic use helps mitigate these effects.

Common fishery anesthetics include tricaine methanesulfonate (MS-222), isoeugenol, eugenol, benzocaine, 2-phenoxyethanol, quinaldine, tetracaine, and bupivacaine. These substances vary in their chemical properties, efficacy, and safety profiles. Depending on their solubility (water-soluble or fat-soluble), they act on the central nervous system to produce reversible sedation or anesthesia. The choice of anesthetic is influenced by factors such as the type of fish, the specific operation, and safety concerns. Figure 1 illustrates their chemical structures, aiding in understanding their molecular characteristics and potential residue behavior.

Appropriate selection and dosing of these anesthetics are critical not only for fish welfare but also for compliance with humane treatment standards and food safety regulations. Moving forward, a deeper exploration of the pharmacological principles of these commonly used anesthetics will help further clarify their mechanisms of action and safety profiles in aquaculture.

## 3. Pharmacological Actions and Safety Assessment

### 3.1. Eugenol

Eugenol (C_10_H_12_O_2_) is a naturally occurring phenolic compound predominantly found in various essential oils, especially clove oil, where it constitutes 85–95% of the active ingredient, along with minor components such as methyl eugenol and isoeugenol [15,16]. Its chemical structure consists of a phenolic ring and an ethylene group, contributing to its water solubility. Eugenol functions as an effective anesthetic by inducing temporary immobility in farmed fish, thereby reducing stress responses during procedures such as transportation, handling, and artificial reproduction. This results in a reduction in physical injuries, including mechanical damage and scale loss, and minimizes the risk of subsequent infections [13]. Additionally, eugenol decreases ammonia excretion in fish by reducing metabolic rates, which enhances water quality and lowers ammonia toxicity [17]. Eugenol also possesses antioxidant and anti-inflammatory properties that help mitigate stress responses during anesthesia [18].

Fishery anesthetics are primarily absorbed through the gills, accumulating in tissues and being eliminated via the same route [19,20]. Studies on eugenol in Pacific white shrimp (*Litopenaeus vannamei*) have demonstrated that, regardless of treatment duration (e.g., 300 mg L^−1^ for 5 min or 10 mg L^−1^ for 24 h), residual eugenol concentrations fall below 2.5 mg kg^−1^ within 24.5 h [21]. This rapid depletion of eugenol residues suggests a low risk of exceeding the FAO/WHO acceptable daily intake (ADI) when shrimp are consumed, indicating its safety for human consumption [21]. Additionally, the half-life of eugenol in seabass varies with water temperature, with faster metabolism observed at higher temperatures (e.g., 2 h at 20 °C vs. over 4 h at 13 °C) [22]. These findings suggest that controlling water temperature can accelerate eugenol metabolism, enhancing residue elimination from fish tissues and supporting food safety.

Although eugenol is effective as an anesthetic, it poses cytotoxic risks to fish at high concentrations. In vitro studies have shown that eugenol can cause developmental abnormalities in zebrafish embryos, including skeletal deformities and pericardial edema, when exposed to concentrations between 0.5 and 2 mg L^−1^ [23]. Tao et al. also found that eugenol exposure delayed zebrafish embryo hatching, reduced body length, and decreased the inflation rate of swim bladders [24]. Moreover, eugenol has been linked to cytotoxicity in various cell lines [25], including mouse fibroblasts [26], rat hepatocytes [27], and human dental pulp fibroblasts [28,29]. In vivo studies also suggest its potential for oral mucosal damage in rats [30,31]. However, despite these cytotoxic risks, studies on rats and other animals have shown no significant toxicity at low concentrations, further supporting the safety of eugenol when used according to recommended guidelines [32].

While eugenol’s safety and efficacy as an anesthetic are well-documented, its regulatory status varies across countries. Eugenol has been approved as a fish anesthetic in several countries, including New Zealand, Japan, Australia, Chile, and Finland [33]. In Japan, the MRL is 50 ng mL^−1^, with a withdrawal period of 7 days for fish and 10 days for crustaceans [7]. The Joint FAO/WHO Expert Committee on Food Additives (JECFA) has set an acceptable daily intake (ADI) of 2.5 mg kg^−1^ for eugenol [34]. However, there are no established MRLs or withdrawal periods for eugenol in the United States and China. The U.S. FDA permits eugenol as a food additive at concentrations up to 1.5 g kg^−1^ but does not approve it for use as a fish anesthetic [35,36]. Given the variability in regulation across regions, it is essential to adhere to local guidelines to ensure the safety of seafood for human consumption.

### 3.2. MS-222

MS-222 (3-aminobenzoic acid ethyl ester methanesulfonate), commonly referred to as tricaine, possesses a chemical structure comprising a benzene ring and an amino acid ester group. It is absorbed into the bloodstream through the gills and skin and is subsequently distributed throughout the body [37]. MS-222 induces anesthesia primarily by blocking sodium ion channels in muscle cells and, to a lesser extent, potassium ion channels in nerve membranes. This mechanism results in a loss of consciousness and a reduction in metabolic activity [38]. The anesthetic efficacy of MS-222 is influenced by biological factors such as fish species and size, as well as environmental conditions including water temperature, pH, and salinity [39]. MS-222 is preferred for its low effective dosage, rapid onset, quick recovery, and minimal side effects, making it a widely used anesthetic in fisheries worldwide [40]. Importantly, MS-222 does not accumulate in fish tissues, leaving minimal residues in muscle, ensuring both its safety and effectiveness. Initially developed as a substitute for cocaine, MS-222 is the only anesthetic approved by the U.S. FDA for aquaculture, although its use is limited to specific fish families, such as *Ictaluridae*, *Salmonidae*, *Esocidae*, and *Percidae* [41]. It is absorbed and eliminated through diffusion across the gill membranes, with no involvement of unmetabolized parent compounds [42]. In fish, MS-222 is primarily metabolized via two pathways: (i) hydrolysis of the ethyl ester to yield m-aminobenzoic acid, which is subsequently N-acetylated, and (ii) N-acetylation of the parent compound followed by hydrolysis [43]. These processes predominantly occur in the liver and gills. In adult rainbow trout, MS-222 and its metabolite, ethyl-meta-acetylaminobenzoate, are excreted through the kidneys. Fish treated with MS-222 in Canada must be held in clean water for 21 days before marketing, with a mandatory 7-day waiting period [11].

While MS-222 is generally safe for use in fish, prolonged exposure in humans, particularly through the consumption of fish with high residual concentrations, can cause skin and respiratory irritation, and in rare cases, retinal damage [44]. However, its high water solubility and rapid metabolism in fish ensure that it typically does not accumulate in edible tissues. When proper withdrawal periods are observed, the residue levels of MS-222 remain sufficiently low to pose minimal risk to human health. Studies have shown that MS-222 significantly reduces stress in fish during anesthesia, thereby enhancing animal welfare in aquaculture operations [45]. Although MS-222 has been shown to be toxic to certain fish species at high concentrations, its high water solubility and the absence of documented human health risks further support its use as a safe anesthetic when the recommended withdrawal period is followed. To ensure food safety, it is crucial to monitor exposure levels, particularly in species that may have heightened sensitivity to MS-222.

MS-222 is approved as a fish anesthetic in several countries, including the United States, Norway, and New Zealand, with withdrawal periods of 21, 21, and 10 days, respectively, before consumption [43]. However, it is not approved in China or the EU, and no official EU standard methods for residue detection have been established. Despite this, MS-222 remains one of the most widely used anesthetics in global aquaculture, with established withdrawal periods and guidelines in regions where it is permitted [43]. The U.S. FDA allows its use in aquaculture with specific restrictions on fish families and requires adherence to withdrawal periods to ensure food safety.

### 3.3. Benzocaine

Benzocaine (C_9_H_11_NO_2_) is a lipophilic compound widely used as a local anesthetic, renowned for its rapid onset, low toxicity, stable efficacy, and prolonged effects. Its chemical structure consists of a benzene ring and an ester group, which enable it to penetrate cell membranes and interact with the nervous system. Benzocaine exerts its anesthetic effect by binding to sodium ion channels, thereby blocking nerve impulse transmission. These properties make benzocaine an ideal anesthetic for a variety of aquatic species, including juvenile fish such as *Colossoma macropomum* and *Aulonocara nyassae*, providing a cost-effective alternative to other anesthetics [46,47]. Once administered, benzocaine is absorbed into fish and metabolized into N-acetylbenzocaine, which is then excreted through the gills, kidneys, and bile [48]. A study by Meinertz et al. on rainbow trout demonstrated that 59.2% of benzocaine residues were excreted through the gills within 3 h, while renal elimination was slower, with only 2.7% excreted through the kidneys after three hours, and 9.0% after 24 h. Bile also contributed to the elimination of 2.0% of the administered dose after 24 h [49]. These findings highlight that the gills are the primary route of elimination, with the kidneys and biliary system serving secondary roles.

Despite its generally low toxicity to fish, excessive human exposure, such as through the consumption of fish with high residual concentrations, can lead to toxicity, including arrhythmias, coma, and pulmonary complications [50]. It may also cause allergic reactions, such as contact dermatitis and hypersensitivity [51]. However, benzocaine’s rapid metabolism and excretion through the gills typically result in low residue levels in fish tissues. When proper withdrawal periods are observed, these residues remain sufficiently low to pose minimal risk to human health. Genotoxicity studies have demonstrated that benzocaine does not possess genotoxic properties, further supporting its safety for human consumption when used according to established guidelines [52].

Benzocaine is subject to regulatory oversight in several countries. In the United States, the FDA has set an import tolerance of 50 μg kg^−1^ for benzocaine residues in fish muscle, with the expectation that residue levels will diminish to minimal concentrations within 24 h after treatment. Regulatory authorities in Australia and New Zealand have set a maximum residue limit (MRL) of 0.05 mg kg^−1^ in finfish, ensuring detectable residues are minimal or below the detection limit [53]. However, China has yet to establish specific regulations for the use of benzocaine in aquaculture, underscoring the need for clearer regulatory frameworks to ensure both food safety and the sustainability of aquaculture practices.

### 3.4. 2-Phenoxyethanol

2-Phenoxyethanol is a colorless, oily liquid that is soluble in water and commonly used as a local anesthetic in aquaculture. Its chemical structure consists of a phenoxy group (C_8_H_10_O_2_) attached to an ethanol group, forming an ether bond between the oxygen atom of the phenoxy group and the ethyl chain of the ethanol group [54]. This molecular configuration gives 2-phenoxyethanol both lipophilic and hydrophilic properties, enabling it to pass through biological membranes efficiently. In fish, it is absorbed through the gills and skin, subsequently transported via arterial blood to the central nervous system, with excretion predominantly occurring through branchial respiration. In rainbow trout, its biological half-life is approximately 30 min [55]. While the exact mechanism of action remains unclear, studies suggest that 2-phenoxyethanol may exert its anesthetic effects by altering the fluidity of neuronal cell membranes, potentially disrupting cellular ion balance and neurotransmission [56]. However, further research is needed to fully understand its anesthetic mechanism.

Although it offers low cost and strong bactericidal properties, 2-phenoxyethanol poses residue concerns, has a prolonged duration of effect, and may be harmful to fish. Compared to anesthetics like MS-222 and eugenol, its use is more restricted, and its efficacy in preserving fish during transport is lower [33]. In the U.S. and European Union, 2-phenoxyethanol is mainly used for non-food fish, aquaculture research, and sedation of ornamental fish during transport [57,58]. The compound is not approved for use in food fish, and experts including Priborsky and Velisek recommend avoiding its application in aquaculture species intended for human consumption due to legal, safety, and environmental considerations [12].

The true anesthetic mechanism of 2-phenoxyethanol in fish is still under investigation. Based on studies in other vertebrates, it is believed to inhibit neural activity in higher regions of the nervous system [59]. Side effects observed include reduced ventilation rate, heart rate, and blood oxygen partial pressure. Exposure in rainbow trout and brown trout has been linked to reductions in red blood cells and platelet counts [60].

Toxicological studies have demonstrated that 2-phenoxyethanol is neither clastogenic nor mutagenic, as evidenced by negative outcomes in the in vivo micronucleus and Ames tests [61]. However, it has been reported to cause toxicity in skin and upper airway tissues [62]. Chronic exposure across multiple species suggests that it may induce hepatotoxicity, renal toxicity, and hemolysis [63]. Notably, a study by Velisek and Svobodova found elevated alanine aminotransferase (ALT) levels in juvenile carp, indicating potential hepatotoxic effects [64].

Currently, no countries explicitly authorize the use of 2-phenoxyethanol for fish intended for human consumption. Its use in food fish is prohibited in the U.S. and the European Union, where it is restricted to non-food fish, aquaculture research, and ornamental fish sedation. Concerns over food safety, legality, and environmental impact limit its broader application in aquaculture.

### 3.5. Diazepam

Diazepam is a long-acting benzodiazepine sedative that acts as a positive allosteric modulator of type-A γ-aminobutyric acid receptors (GABAARs), enhancing the effect of γ-aminobutyric acid (GABA) and reducing neuronal activity [65]. It is commonly used to treat neurological disorders such as epilepsy, anxiety, and sleep disturbances. In aquaculture, diazepam reduces the metabolic rate in fish, promoting growth, alleviating stress, and improving survival rates. It also induces a schooling effect in fish when added to their feed. However, the use of diazepam at any stage can lead to its residues in fish, which are persistent and can transfer through the food chain to humans. Studies have shown that diazepam residues in freshwater fish such as *Parabramis pekinensis* and *Carassius auratus* can reach concentrations of 0.5–118.6 μg kg^−1^, with a detection rate as high as 26.8% [66]. Despite its beneficial effects in aquaculture, the persistence of diazepam residues remains a significant concern.

Although the pharmacokinetics of diazepam in aquatic species has not been extensively studied, research in mammals suggests that diazepam is well absorbed after oral administration, reaching peak plasma concentrations within 30 to 90 min [67]. Its metabolism is influenced by factors such as age, gender, liver disease, and genetic variations, particularly those affecting cytochrome P450 enzymes [68]. In pregnant women, diazepam rapidly crosses the placenta, leading to prolonged sedation in newborns. In fish, its absorption through the gills and skin, followed by distribution to the central nervous system and excretion through branchial respiration, suggests similar patterns of pharmacokinetics, though more research is needed to confirm this in aquatic species.

The use of diazepam in aquaculture and during fish transportation is prohibited; however, residues have still been detected in animal products. A survey conducted by the Hunan Provincial Department of Agriculture identified residues above permissible limits in carp, with 4.9% of 286 samples from various species testing non-compliant [69]. These findings highlight ongoing concerns regarding human exposure through the food chain.

Diazepam residues in fish can be harmful if consumed by humans. Ingesting contaminated fish may cause symptoms such as fatigue, drowsiness, ataxia, and mental confusion. In severe cases, it may lead to coma, arrhythmias, or carcinogenic effects [70,71]. The persistence of diazepam in fish tissues and its potential transfer through the food chain underscore the health risks associated with its illegal use in aquaculture, particularly when withdrawal periods are not observed.

Many countries prohibit the use of diazepam in food animals, due to concerns over food safety and human health. In China, this restriction was formalized by the Ministry of Agriculture through Announcement No. 193 (2002) [72], which listed sedative-hypnotic drugs, including diazepam, as banned substances in food-producing animals. The current national standard, Maximum Residue Limits of Veterinary Drugs in Food (GB 31650-2019) [73], further stipulates that while diazepam may be administered to non-food animals, residues are not permitted in edible animal products. Similarly, the European Union’s Food and Feed Safety Law and Residue Regulation (Regulation (EC) No 470/2009) [74] prohibit diazepam in food products, especially in raw materials like meat, dairy, and poultry [75]. Although the Codex Alimentarius Commission (CAC) has not set specific standards for diazepam, it mandates that all drug residues in food must be below safe levels for human consumption. Despite these prohibitions, illegal use of diazepam in aquaculture persists, leading to contamination of fish and seafood products, which are subject to regulatory actions aimed at preventing human exposure.

### 3.6. Quinaldine

Quinaldine (2-methylquinoline) is an alkaloid anesthetic with antibacterial and antipyretic properties, widely used as a precursor for the synthesis of various pharmaceuticals, including biocides and bactericides [76]. Its chemical structure consists of a quinoline ring system, a bicyclic structure comprising a benzene ring fused to a pyridine ring, with a methyl group (-CH_3_) attached to the second position of the quinoline nucleus. The presence of a nitrogen atom in the pyridine ring contributes to its basicity, allowing quinaldine to interact with biological systems through ion exchange and receptor binding. This molecular configuration imparts lipophilicity to quinaldine, enabling efficient absorption through cellular membranes. Since the 1950s, quinaldine has been investigated as a fish anesthetic and has since been widely applied in fisheries for processes such as capture, transportation, sampling, and measurement [76]. At low doses, quinaldine effectively anesthetizes fish species such as Rohu (*Labeo rohita*) and Silver Carp (*Hypophthalmichthys molitrix*), inducing prolonged anesthesia lasting up to 6 h. This extends the window of sedation, reduces stress, and prevents bacterial growth, thereby improving fish survival during transport [77]. Furthermore, combination strategies have shown enhanced efficacy; for instance, co-administration of quinaldine with diazepam significantly reduced induction time and excitability in seabream compared to quinaldine alone [78].

However, systematic pharmacokinetic studies of quinaldine in aquatic species remain limited. Current knowledge is largely based on observations of anesthetic onset and duration, with low doses producing several hours of sedation [77]. Data on absorption, distribution, metabolism, and excretion in fish remain scarce, leaving the residue profile of quinaldine insufficiently characterized.

Research on the toxicological effects of quinaldine is also limited. Existing studies suggest that low concentrations are generally safe for fish anesthesia, but its environmental impact raises concern. Quinaldine-containing wastewater is resistant to degradation and may pose long-term ecological risks [79]. Additionally, the potential for bioaccumulation in fish and subsequent risks to human consumers has not been fully evaluated, highlighting the need for further investigation.

At present, no international or national authority explicitly authorizes quinaldine for use in food fish. Its application is generally restricted to research and fish transport. Due to the lack of clear residue data, toxicological evidence, and food safety assessments, regulators remain cautious, and quinaldine is not approved for aquaculture use in species intended for human consumption.

## 4. Considerations on the Applicability of Commonly Used Anesthetics in Aquaculture

In aquaculture, the effectiveness of anesthetics is influenced by both biological factors (such as species, gender, age, life stage, health status, and stress response) and abiotic factors (including salinity, pH, oxygen content, and water temperature) [80,81,82]. Generally, as anesthetic concentration increases, the induction time decreases; however, the time required for full recovery significantly increases [21,82]. Higher water temperatures lead to increased oxygen consumption and breathing frequency in aquatic animals, as well as enhanced gill water exchange, which accelerates anesthetic absorption and shortens anesthesia duration [83,84]. Conversely, under lower temperature conditions, both induction and recovery times are prolonged [85]. Furthermore, at the same anesthetic concentration, longer exposure times and greater body weight are associated with longer recovery times [86,87].

When selecting an anesthetic for aquaculture species, it is important to consider the species-specific and environmental factors. Eugenol, for example, is suitable for a variety of aquaculture species, including freshwater tropical fish (such as guppy), large freshwater/marine fish, and crustaceans (such as shrimp) [88]. Eugenol has good induction and recovery effects for crustaceans, with low cost and high safety. For cold-water species (such as rainbow trout, salmon, etc.) [21,82], MS-222 is commonly used, especially effective at lower temperatures [89]. However, MS-222 is ineffective for most farmed crustaceans [90]. For tropical fish species (such as tilapia and guppy), eugenol is the preferred anesthetic, particularly at warmer water temperatures [13]. For shrimp (such as *Litopenaeus vannamei*), both eugenol and benzocaine are effective, but the dose should be adjusted based on the size of the shrimp, with sufficient metabolic clearance time ensured to avoid adverse reactions [82,91].

Ghanawi et al. evaluated the efficacy of four anesthetics (eugenol, benzocaine, 2-phenoxyethanol, MS-222) on juvenile *Siganus rivulatus* and determined their optimal concentrations. The study showed that all anesthetics were effective for *Siganus rivulatus*, with 2-phenoxyethanol being the most suitable for handling and transportation, as it did not induce abnormal swimming behavior during the 24 h anesthetic bath, had a short recovery time, and showed no observed mortality [92]. However, 2-phenoxyethanol may pose a potential hazard to aquaculture workers, limiting its use in edible fish [93,94]. Both MS-222 and eugenol can also be used for transportation, but some fish exhibited abnormal swimming behavior during anesthesia and recovery. Benzocaine was considered the least suitable anesthetic, as it required a longer recovery period and caused mortality during anesthesia and recovery [92].

Munday and Wilson reported that quinaldine was the most effective anesthetic for the coral reef fish *Pomacentrus amboinensis*. Compared to benzocaine, MS-222, 2-phenoxyethanol, and eugenol, quinaldine caused complete loss of balance, with faster and more stable anesthetic effects [77,95]. However, the use of diazepam in aquaculture presents certain issues. When diazepam is added to feed, it stimulates fish to eat, thereby increasing catchability, but this also introduces the potential risk of diazepam residue [96]. Moreover, diazepam has toxic effects on lipid peroxidation, biochemical, and oxidative stress indicators in the liver and gill tissues of African catfish [97]. Qi et al. found that diazepam is quickly absorbed, widely distributed, and slowly eliminated after oral administration in *Carassius auratus*, with a half-life of 619.31 h in muscle and skin tissues, requiring at least 70 days for the concentration to fall below the quantification limit [98].

In summary, good aquaculture practices, including water quality control, appropriate withdrawal periods, and effective detection methods, are crucial to ensuring the efficacy of anesthetics and food safety.

## 5. Analytical Methods for Residue Detection

### 5.1. Instrument Detection

Monitoring residual anesthetic levels in aquatic products is essential for ensuring food safety and regulatory compliance. However, detection is challenging due to the complex composition of aquatic products, which contain proteins, fats, and other components that can interfere with accurate quantification. Moreover, anesthetic residues, such as those from eugenol, MS-222, benzocaine, 2-phenoxyethanol, Diazepam and their metabolites, are usually present at low concentrations. These residues and metabolites may exhibit toxic or pharmacological effects, necessitating highly sensitive and reliable detection methods. The detection is further complicated by their diverse chemical properties, including volatility, solubility, and polarity. Each anesthetic requires a specific detection protocol tailored to its characteristics, necessitating flexible methodologies to account for variations in chemical behavior. Pre-treatment steps, such as extraction and purification, are crucial for minimizing interference and ensuring accurate quantification. Common pre-treatment methods include Soxhlet extraction (SE) [99], liquid–liquid extraction (LLE) [100], solid-phase extraction (SPE) [101], headspace solid-phase microextraction (HS-SPME) [102], Quick, Easy, Cheap, Effective, Rugged and Safe (QuEChERS) [103], and molecular imprinting techniques (MIT) [104]. In summary, detecting anesthetics in aquatic products requires advanced analytical instruments, coupled with complex sample preparation workflows, to ensure accuracy and sensitivity. Each method must be optimized based on the chemical properties of the anesthetic, making the process particularly challenging. Table 2 summarizes key analytical parameters for the instrumental analysis of anesthetics in aquatic products.

#### 5.1.1. GAS Chromatography (GC, GC-MS)

GC and GC-MS are widely preferred for anesthetic residue analysis due to their high sensitivity, effective separation capabilities, and precise quantification of trace anesthetics. Among the anesthetics, eugenol, MS-222, and benzocaine are particularly suited for GC-MS analysis. Li et al. developed a new pretreatment method for eugenol in fish samples using the Stable Isotope Dilution Assay (SIDA) and SPE, effectively minimizing matrix effects during GC-MS analysis. The combined SIDA-SPE-GC-MS/MS approach demonstrated accuracy and precision that meet bioanalytical assay requirements. Among the evaluated techniques, Vortex-Assisted Liquid–Liquid Microextraction (VALLME) combined with HS-SPME pretreatment achieved the lowest detection limits, making it highly effective for eugenol detection [119]. Liang et al. established an HS-SPME method for eugenol extraction from fish, ensuring GC-MS stability while incorporating VALLME to reduce matrix effects and enhance sensitivity. The optimized pretreatment demonstrated good repeatability, linearity, and sensitivity, making it suitable for long-term GC-MS analysis [105]. Huang et al. were the first to employ a Multiplug Filtration Cleanup (m-PFC) method (Figure 2A), derived from QuEChERS, in combination with Gas Chromatography-Orbitrap Mass Spectrometry (Orbitrap GC-MS) for the determination of six eugenol anesthetics in aquatic products [110]. This rapid pretreatment method exhibited strong resistance to matrix interference, enabling accurate detection in large sample quantities.

For MS-222, a water-soluble anesthetic, GC can be employed for detection [112]. As shown in Figure 2B, a biocompatible PDMS fiber head extraction technique demonstrates balanced in vitro and in vivo extraction of five anesthetics from tilapia, highlighting its efficiency in complex biological matrices. This approach offers a short analysis time and prevents analyte loss due to degradation or sample storage [120].

Abreu et al. developed and compared SPME and single-drop microextraction (SDME) techniques for detecting 2-phenoxyethanol residues in fish fillets, using a central composite design (CCD) to enable accurate assessments with minimal sample volume. Both methods demonstrated good precision, with SDME achieving detection and quantification limits of 0.2 μg mL^−1^ and 0.62 μg mL^−1^, respectively, while SPME achieved limits of 0.18 and 0.56 μg mL^−1^. At anesthetic concentrations of 450–1050 μg mL^−1^, the elimination times for 2-phenoxyethanol were 12 h for SDME and 24 h for SPME, indicating that both techniques are feasible for residue analysis [102].

#### 5.1.2. Liquid Chromatograph (LC or HPLC)

Unlike GC, liquid chromatography (LC) and high-performance liquid chromatography (HPLC) do not require analytes to be volatilized, making them particularly suitable for detecting non-volatile or thermally unstable anesthetics. In complex matrices such as aquatic products, HPLC offers superior precision and repeatability in quantification, establishing it as a more reliable method for quantitative analysis [123].

Scherpenisse and Bergwerff investigated various extraction columns for detecting MS-222 residues in the tissues of three fish species. Their findings revealed that the C-18 column achieved a higher fortified recovery rate, complying with FDA and Canadian standards for MS-222 [124]. For the rapid determination of diazepam and its main metabolites in fish samples, Li et al. employed primary secondary amine (PSA) and multi-walled carbon nanotubes (MWCNT) as QuEChERS sorbents, coupled with high-performance liquid chromatography-electrospray ionization-tandem mass spectrometry (HPLC-ESI-MS/MS). The study showed that PSA exhibited superior extraction efficiency than MWCNT, effectively removing interfering substances. This approach yielded excellent recovery and an acceptable relative standard deviation (RSD), with a limit of quantification (LOQ) of 2.5 μg kg^−1^ [70]. Xie et al. developed a stable isotope dilution assay coupled with HPLC-tandem mass spectrometry to quantify MS-222 levels in grass carp, utilizing a synthesized stable isotope-labeled Tricaine-D_5_ to enhance accuracy and precision. This method demonstrated high adsorption capacity for MS-222, providing a rapid and accurate approach for detecting trace levels in aquatic products [113]. In another study, Xia et al. synthesized uniform magnetic covalent organic framework (MCOF) microflowers by embedding aldehyde-functionalized Fe_3_O_4_ nanoparticles into a covalent organic framework (COF) under mechanical stirring at room temperature. Under optimal conditions, these MCOF microflowers exhibited rapid adsorption (10 min) and high extraction efficiency (over 84.02%) for eugenol anesthetic (Figure 2C). The developed MSPE-HPLC-UV method demonstrated high precision and accuracy, enabling effective quantification of three eugenol anesthetics in tilapia, shrimp, and crab samples [121].

To achieve selective recognition of diazepam, Aitor et al. used magnetic stir bars with an activated polytetrafluoroethylene (PTFE) surface, onto which a molecularly imprinted polymer (MIP) monolith was covalently bound. This MIP system enhanced extraction performance (Figure 2D). In addition, two analytical techniques have been proposed for determining diazepam in water extracts: ultra-high-performance liquid chromatography coupled with mass spectrometry (UHPLC-MS/MS) and ion mobility spectrometry (IMS). Among these, UHPLC-MS/MS demonstrated greater sensitivity, with an limit of detection (LOD) of 1.2 ng L^−1^, compared to IMS (LOD of 1.2 μg L^−1^) [122].

In summary, to effectively detect anesthetic residues, it is essential to match the anesthetic type with the appropriate pre-treatment method. GC and GC-MS are among the most widely used methods for anesthetic residue analysis due to their high sensitivity, effective separation capabilities, and precise quantification of trace anesthetics. These methods are especially effective for detecting volatile or semi-volatile anesthetics like eugenol and MS-222. LC and HPLC are particularly suitable for anesthetics like 2-phenoxyethanol and benzocaine, which are commonly used in aquaculture but are more challenging to analyze using GC. GC-MS and LC-MS/MS remain the most stable and sensitive techniques for residue detection, partly due to their ability to stably analyze anesthetic and anesthetic metabolites, which are crucial for understanding the full scope of anesthetic exposure and residue presence. However, the development of effective pre-treatment extraction methods and the use of appropriate pre-treatment materials continue to be important areas of research to ensure the reliability and accuracy of these techniques.

### 5.2. Rapid Detection

Although the instrument exhibits high analytical accuracy, its applicability for the rapid detection of large quantities of on-site samples is limited due to labor-intensive pre-processing and substantial resource consumption, requiring specialized and costly technical personnel [125]. In contrast, rapid analytical methods for detecting fishery anesthetics offer significant advantages, including efficiency, sensitivity, and simplified pre-processing. These methods enable the rapid detection of on-site samples, addressing the limitations associated with conventional instrumental detection [126]. Immunoassay and electrochemical detection methods are currently utilized to rapidly determine anesthetic residues in fish.

#### 5.2.1. Immunoassays

Immunoassay is a rapid analytical technique that relies on the specific binding reaction between antigens and antibodies. Immunoassay-based detection of fishery anesthetics provides rapid and sensitive screening alternatives to instrument-based methods. Common immunoassay methods include enzyme-linked immunosorbent assay (ELISA), colloidal gold immunochromatographic assay (GICA), and fluorescent immunoassay (FIA) [127].

ELISA is a versatile analytical technique capable of both qualitative and quantitative detection, relying on the immobilization of antigens or antibodies on a solid carrier, followed by colorimetric detection and analysis [128]. This technique is highly valued for its sensitivity, simplicity, minimal sample pretreatment requirements, and capacity to efficiently process large numbers of small-volume samples [129]. GICA is an innovative immunolabeling technique that utilizes colloidal gold as a tracer for labelling monoclonal antibodies [130]. In this method, a competitive reaction occurs between the target analyte in the sample and the antigen on the test line, producing visible chromogenic results that can be observed with the naked eye [131]. FIA integrates the specificity of immune responses with the sensitivity of fluorescence techniques, offering a highly effective tool for detection and analysis [132].

Lateral flow immunochromatography is a cost-effective, user-friendly, and time-efficient method that offers sufficient sensitivity, accuracy, and specificity [133,134]. For instance, Shen et al. developed a colloidal gold-based immunoassay for detecting four eugenol compounds (EUGs) in water, with a detection range of 5–100 µg mL^−1^ [130]. As shown in Figure 3A, the lateral flow immunochromatographic strip (LF-ICS) enables portable and rapid MS-222 detection, with visual detection and cut-off limits of 0.1 µg mL^−1^ and 1 µg mL^−1^, respectively [135]. Subsequent research by the group led to the development of a four-layer immunochromatographic assay (Qua-ICS) based on colloidal gold, which incorporated four highly sensitive monoclonal antibodies (mAbs) to simultaneously detect 11 anesthetic residues in fish within 10 min. The method exhibited detection ranges of 3.3–10, 11–222, 100–2000, 0.37 and 3.3, and 111–10,000 µg kg^−1^ (Figure 3B). Quantitative analysis was performed using a portable strip reader, with detection ranges of 0.15–2.6, 6.3–677, 0.13–2.8, and 83–1245 µg kg^−1^ for procaine, eugenol, bupivacaine, and tetracaine, respectively. Compared to traditional single ICS methods, this multiplex ICS enables the simultaneous detection of 11 anesthetic residues within 10 min, providing a more comprehensive screening approach for anesthetic residues in fish [136].

GICA is primarily suitable for qualitative or semi-quantitative detection. However, for precise quantification, fluorescent immunoassays have been developed [139,140]. Integrating nanomaterials in ELISA enhances the fluorescence signals, improving detection efficiency. Ratiometric fluorescence detection further advances this technique by quantifying analytes based on the intensity ratio of emissions at two different wavelengths [141,142]. This method not only preserves the high sensitivity of fluorescence detection but also incorporates an internal calibration mechanism, effectively mitigating erroneous signals caused by external environmental factors.

For example, our group developed a monoclonal antibody specific to MS-222 and utilized manganese dioxide nanosheets to mediate fluorescence reactions, generating two distinct fluorescence signals for detecting MS-222 residues in aquatic products (Figure 3C). The resulting ratiometric fluorescence ELISA (RF-ELISA) achieved a limit of detection of 0.28 ng mL^−1^ in buffer, which is 376 times lower than that of conventional colorimetric ELISA. In shrimp and tilapia samples, the LODs were 2.8 ng g^−1^ and 5.6 ng g^−1^, respectively [125].

Additionally, our group developed an expandable ratiometric fluorescence sensing platform (Figure 3D). The core mechanism of this platform is based on the alkaline phosphatase (ALP)-catalyzed hydrolysis of ascorbic acid 2-phosphate (AAP), releasing ascorbic acid (AA), which inhibits the dopamine (DA)-mediated synthesis of luminescent polydopamine (PDA). Simultaneously, PDA effectively quenches the fluorescence of carbon dots (CDs), generating a distinct ratiometric fluorescence signal. This multifaceted system enables the RF sensor to achieve ultra-sensitive detection of ALP activity, with a detection limit as low as 0.01 mU L^−1^ for benzocaine [137]. Similarly, our group employed red-emitting gold nanoclusters (Au NCs) as fluorescence probes to quench the fluorescence of 2,3-diaminophenazine (DAP), the oxidation product of o-phenylenediamine (OPD) catalyzed by horseradish peroxidase (HRP) (Figure 3E). Utilizing broad-spectrum monoclonal antibodies against EUGs and enzyme-labeled secondary antibodies, we recently developed a ratiometric fluorescence immunoassay with sensitivity at the pg mL^−1^ level [138]. This approach offers a rapid and efficient screening method for eugenol-based anesthetics in aquatic products while also serving as a reference for developing immunoassays targeting other small molecule contaminants, contributing to food safety and public health.

#### 5.2.2. Electrochemical Sensor

Electrochemical sensor leverages the electrochemical properties of target analytes to convert their chemical quantities into electrical signals [143]. As a rapid detection technology, it offers key advantages such as ease of operation, high sensitivity, and strong potential for miniaturization. Furthermore, the superior electrochemical performance of novel nanomaterials in electrode applications has accelerated advancements in electrochemical detection [144]. These attributes have contributed to its widespread application in diverse fields, including food safety and environmental monitoring.

Rafael et al. developed a batch injection analysis (BIA) system utilizing screen-printed carbon electrodes for the rapid and precise quantification of the anesthetics benzocaine and MS-222 in fresh fish fillets [145]. This method exhibits high sensitivity, enabling over 300 injections per hour, with low detection limits and excellent reproducibility. In addition, it effectively minimizes matrix interference in the samples, enhancing analytical accuracy. Regarding MS-222, Cai et al. developed a nanoporous gold (NPG) electrochemical sensor using a simple and efficient one-step corrosion method with concentrated nitric acid (Figure 4A). Their study demonstrated that the NPG sensor exhibits high sensitivity, a broad linear range, and reasonable recovery rates, along with user-friendly operation, making it well-suited for rapid on-site sample testing [146]. Moreover, as shown in Figure 4B, Shi et al. developed a platinum nanoparticle/raspberry-like SiO_2_-modified glassy carbon electrode (Pt NPs@RL-SiO_2_/GCE), which demonstrated excellent electrocatalytic activity for the simultaneous detection of eugenol and methyleugenol [147]. Similarly, Chen et al. constructed a chitosan-reduced graphene oxide/multimetal oxide/poly-L-lysine (CS-rGO/P_2_Mo_17_Cu/PLL) modified electrode via electrodeposition, achieving a detection limit of 0.4490 μM. This electrode demonstrated effective applicability for detecting EU residues in the kidneys, liver, and muscle tissues of freshwater bass [148] (Figure 4C).

Finally, several innovative electrochemical sensor electrodes have recently been developed for the sensitive voltammetric quantification of benzocaine. For example, Pysarevska et al. were the first to utilize a miniaturized boron-doped diamond thick film electrode as an advanced electrochemical sensor [149]. Meanwhile, Wang et al. leveraged the remarkable properties of β-cyclodextrin to fabricate a carbon black/β-CD nanocomposite electrode [150]. However, current research on the detection of fishery anesthetics remains limited, underscoring the need for further exploration in this field. Advancing the application of electrochemical sensors for the rapid detection of fishery anesthetics represents a promising direction for future research.

## 6. Conclusions

Fishery anesthetics, such as eugenol, MS-222, and benzocaine, are essential tools in aquaculture, providing effective methods for reducing stress and improving animal welfare during farming operations. However, significant challenges remain due to the lack of harmonized regulations across regions and concerns about the accumulation of residues in edible tissues, which can impact both food safety and consumer confidence. The illegal use of certain sedatives, such as diazepam, further complicates the situation, raising concerns about bioaccumulation and long-term food safety risks.

Looking ahead, several areas of research are critical to advancing the field. First, the development of new, eco-friendly anesthetic formulations that are both low in toxicity and capable of rapid biodegradation is a key priority. Such formulations would reduce the risk of residual buildup in aquatic organisms, addressing both environmental and food safety concerns. Second, advancements in multi-residue detection technologies, such as the development of more sensitive biosensors and point-of-care testing methods, will enable faster, more efficient monitoring of anesthetic residues in aquatic products.

Moreover, the adoption of a One Health approach—integrating human, animal, and environmental health considerations—is essential for ensuring comprehensive safety in aquaculture. Globally harmonized regulatory frameworks will play a pivotal role in reducing trade barriers, facilitating safer and more sustainable aquaculture practices across different regions. Equally important is the promotion of responsible aquaculture practices through targeted training programs for producers, which will help minimize anesthetic use and encourage more sustainable farming methods.

## Figures and Tables

**Figure 1 foods-14-03928-f001:**
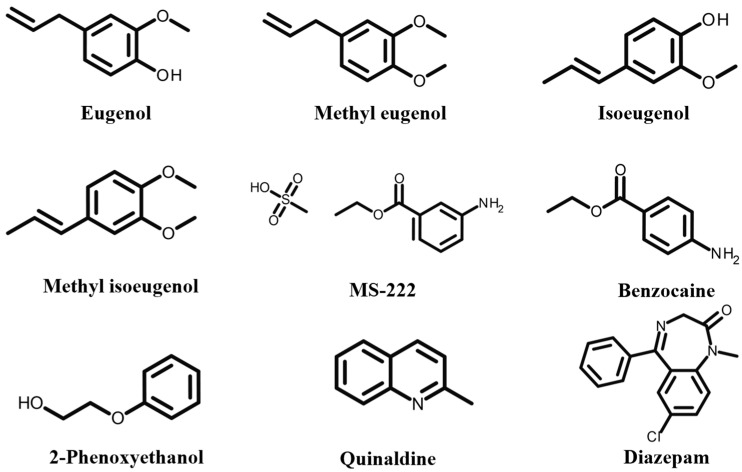
The chemical structure of the anesthetics.

**Figure 2 foods-14-03928-f002:**
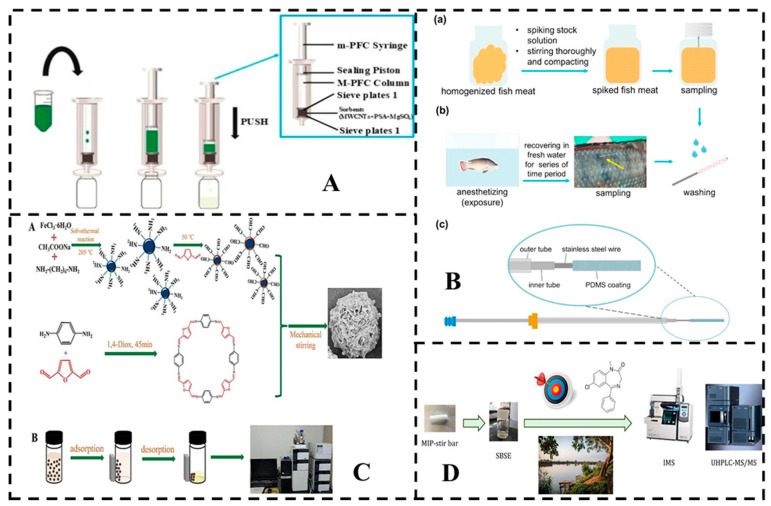
Instrument detection of fishery anesthetics. (**A**) Purification procedure diagram of *m*-PFC column. Reprinted with permission from Huang, 2021 [110]. (**B**) Biocompatible PDMS fiber heads were used for balanced extraction of five anesthetics from tilapia in vitro and in vivo. The procedure of in vitro and in vivo experiment. (a) In vitro experiment was conducted in spiked homogenized fish dorsal meat; (b) It shows the procedure of elimination experiment (in vivo sampling), and the experiments on real samples purchased from local market were the same excluding the anesthetizing and recovering steps. (c) The schematic diagram of the commercial SPME fiber assembly fixed with a home-made PDMS fiber. After extraction, the PDMS fiber was inserted and fixed in the inner tube, which would be protected by the outer tube when inserting into the injection port, then pushed out and exposed. Reprinted with permission from Huang, 2017 [120]. (**C**) Schematics for the synthesis of MCOF microflowers (A) and their MSPE application for eugenol anesthetics enrichment for HPLC analysis (B). Reprinted with permission from Xia, 2023 [121]. (**D**) Adsorption and extraction of diazepam from natural water using a molecularly imprinted polymer stir bar, combined with analysis via MIP stir-bar sorptive extraction (SBSE) and UHPLC-MS/MS. Reprinted with permission from Sorribes-Soriano, 2022 [122].

**Figure 3 foods-14-03928-f003:**
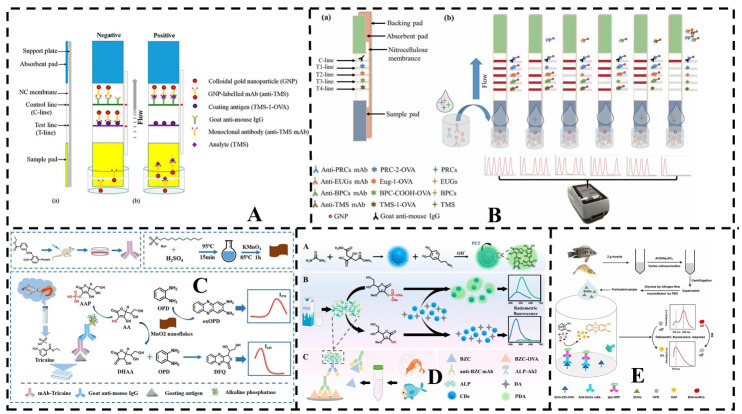
Immunoassay analysis of fishery anesthetics. (**A**) Schematic illustrations of the LF-ICS. (**a**) The structure of the LF-ICS, (**b**) The principle of the LF-ICS test for TMS. Reprinted with permission from Lei, 2020 [135]. (**B**) The (**a**)construction and (**b**) principle of the colloidal gold-based quadruplex immunochromatographic (Qua-ICS) assay. Reprinted with permission from Lei, 2023 [136]. (**C**) Principle of the MnO_2_ nanoflake-based ratiometric fluorescent immunoassay for MS-222. Reprinted with permission from Liang, 2023 [125]. (**D**) Schematic illustration of controlled dopamine formation on carbon dots for ultrasensitive detection of alkaline phosphatase and ratiometric fluorescence immunoassay of benzocaine (A–C). Reprinted with permission from Lin, 2023 [137]. (**E**) Schematic illustration of the ratiometric fluorescence immunoassay for determination of EUGs. Reprinted with permission from Luo, 2024 [138].

**Figure 4 foods-14-03928-f004:**
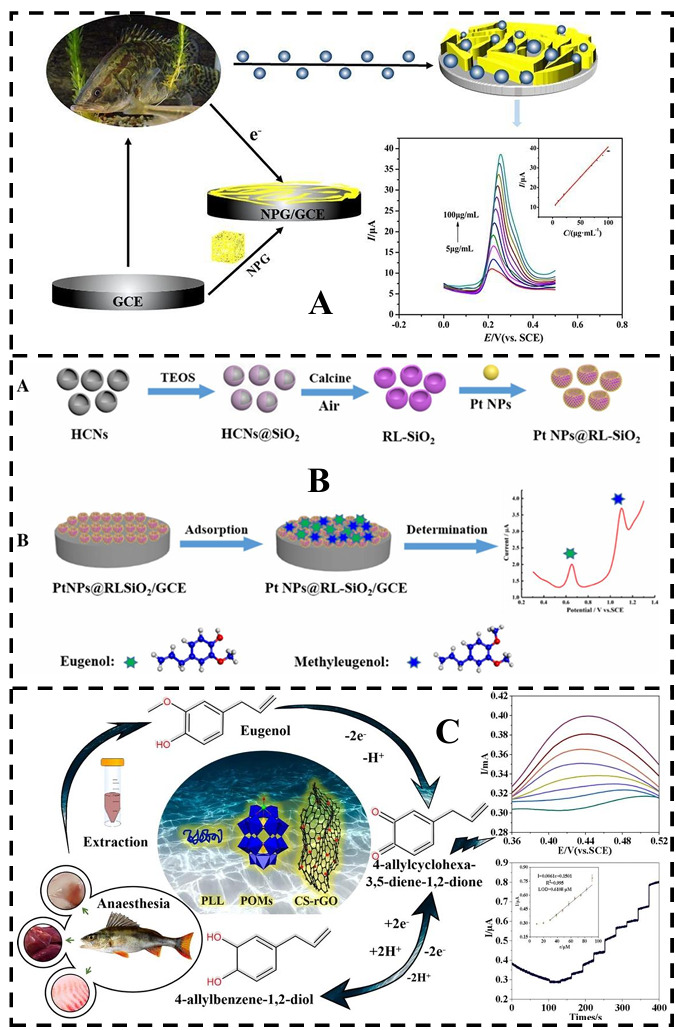
Electrochemical detection of fishery anesthetics. (**A**) Schematic diagram of a nanoporous gold electrochemical sensor for detecting MS-222 Reprinted with permission from Cai, 2021 [146]. (**B**) The synthesis (A) and application schematic diagram (B) of Pt NPs@RL-SiO_2_. Reprinted with permission from Shi, 2021 [147]. (**C**) Schematic diagram of rapid detection of eugenol in perch using an electrochemical method with transition metal-substituted polyoxometalates. Reprinted with permission from Chen, 2023 [148].

**Table 1 foods-14-03928-t001:** Withdraw period and maximum residue limit standards for different anesthetics in some countries/organization.

Anesthetic	MRL		Withdrawal Period	Reference
Eugenol	New Zealand	100 ng mL^−1^	—	[6]
	Japan	50 ng mL^−1^	7 d	[7]
Isoeugenol	European Conformity (CE)	6 mg kg^−1^	—	[8]
AQUI-S	Australia and Chile	—	0d	[9]
MS-222	FDA	1 µg mL^−1^	21 d	[10]
	Canada	—	7 d	[11]

**Table 2 foods-14-03928-t002:** Representative analytical parameters for the analysis of anesthetics using instrumental methods in aquatic products.

Methods	Anesthetic	Sample	Pretreatment	Linearity	Repeatability/Reproducibility	Detection Limits	Reference
GC-MS	eugenol	fish back, fish belly, and fish tail	VALLME/HS-SPME	15.0–750.0 μg kg^−1^	RSD < 20%, *n* = 3	0.5 μg kg^−1^	[105]
	eugenol	carp muscle tissues	SPE	5.0–500.0 μg L^−1^	RSD < 12%, *n* = 6	2.5 μg kg^−1^	[7]
	Isoeugenol	shrimp, tilapia, and salmon	Headspace solid-phase microextraction	0–160 ng g^−1^	RSD 5–13%, *n* = 9	below 15ng g^−1^	[106]
GC-MS/MS	diazepam	water samples	dispersive solid-phase microextraction	10–1000 ng mL^−1^	RSD = 6%, *n* = 5	3 ng mL^−1^	[107]
	eugenol, isoeugenol‚ and methyleugenol	groupers	SPE	5–500 μg L^−1^	RSD 2.18–15.5%, *n* = 4	eugenol 0.4 μg kg^−1^, isoeugenol 1.2 μg kg^−1^ ‚ and methyleugenol 0.2 μg kg^−1^	[33]
	2-Phenoxyethanol	rainbow trout	SPME	0.1–250 mg kg^−1^	RSD 3–11%, *n* = 5	0.03mg kg^−1^	[108]
GC-IT-MS/MS	eugenol	mandarin	QuEChERS	5–1000 μg L^−1^	RSD 1.82–9.74%, *n* = 6	5.0 μg/kg	[109]
Orbitrap GC-MS	Eugenol, methyleugenol, isoeugenol, methyl isoeugenol, eugenol acetate, and acetyl isoeugenol	prawns	m-PFC column	0.001–0.1 μg mL^−1^	RSD 1.2–7.5%, *n* = 6	2–10 μg kg^−1^	[110]
HPLC-MS/MS	MS-222	carp and eel	QuEChERS	2–1000 μg L^−1^	RSD < 6%, *n* = 3	2.5 μg kg^−1^	[111]
		/	SPE	0.05–10 μg L^−1^	RSD < 9.36%, *n* = 5	0.01 μg/L	[112]
		finfish	extracted with acetone using a tissue homogenizer, followed by derivatization with dansyl chloride	2.5–40.0 ng g^−1^	RSD 2.6–8.0%, *n* = 6	0.2–1 μg kg^−1^	[100]
		marine fish and freshwater fish	isotope dilution assay	2.0–200.0 μg L^−1^	inter- and intra-assay relative standard deviations (RSD values) were 0.39–3.01 and 0.85–2.77%	1 μg kg^−1^	[113]
LC-MS/MS	diazepam	fish and shrimp tissue	C_18_ cartridge solid-phase extraction	0.05–5ng mL^−1^	RSD < 4.9%, *n* = 3	0.01 μg kg^−1^	[114]
LC	AQUI-S^®^ 20E (eugenol)	standard water containing fish feed	SPE	5–500mg L^−1^	RSD < 0.7%, *n* = 3	0.0011 mg L^−1^	[115]
HPLC-QTRAP-MS/MS	tricaine, tetracaine, and bupivacaine	fish samples	QuEChERS	1.0–50.0 μg L^−1^	RSD < 15%, *n* = 3	2.0 μg kg^−1^	[116]
PGD-IMS/LC-MS/MS	MS-222	fish-raising water samples	m-PFC	0.005–0.2 mg L^−1^	PGD-IMS RSD 6.9–10.3%, *n* = 5 LC-MS/MS RSD 1.3–3.4%, *n* = 5	6 μg kg^−1^/0.6 μg kg^−1^	[11]
LC-QLIT-MS/MS	eugenol	fish samples	dispersive solid-phase extraction (DSPE)	1–100 μg kg^−1^	RSD 1.9–8.9%, *n* = 6	0.0–0.4 μg kg^−1^	[117]
HPLC-UV	Diazepam	water samples	Dispersive micro solid phase extraction	0.3–450 ng mL^−1^	RSD 3.42–3.75%, *n* = 3	0.09 ng mL^−1^	[118]

## Data Availability

No new data were created or analyzed in this study. Data sharing is not applicable to this article.

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
