# Peer review of "Fishery Anesthetics in Aquaculture Products: Safety Concerns and Analytical Methods"

_foods, 2025, doi:10.3390/foods14223928_

Round 1

Reviewer 1 Report

Comments and Suggestions for Authors

I am a professor of food safety, and seafood is one of my favorite areas of study. I read the article with great pleasure and thoroughly enjoyed it. My compliments to the authors!
I found some typos in the text, which I have highlighted in my draft, which I have attached as a PDF.
In my revision, I also highlighted two points that I encourage the authors to consider and, if necessary, update their writing.
In any case, a very interesting article, with very well-crafted illustrations.

Comments on the Quality of English Language

The proposed article is written in correct and understandable English. I have merely pointed out some typos I found in the text.

Author Response

Reviewer 1:

1.insert a space before playing, please.

Response: Thank you for your feedback. We will insert a space before "playing" as requested. We also checked the entire text to avoid such mistakes.

  1. The cited reference indicates that the species is Litopenaeus vannamei. I propose citing it in the article's text as well, for completeness. Thank you.

Response: Thank you very much. We update the manuscript to include the species Litopenaeus vannamei in the relevant section for completeness, ensuring that it aligns with the cited reference. Here is the revised sentence:

Studies on eugenol in Pacific white shrimp (Litopenaeus vannamei) have demonstrated that, regardless of treatment duration (e.g., 300 mg L⁻¹ for 5 minutes or 10 mg L⁻¹ for 24 hours), residual eugenol concentrations fall below 2.5 mg kg⁻¹ within 24.5 hours [15].

  1. Please insert a space after the preceding dot.

Response: Thank you for your feedback. We will insert a space before " Additionally, " as requested. We also checked the entire text to avoid such mistakes.

  1. Move the dot closer to sensitivity, please.

Response: Thank you very much. We moved the dot closer to the word "sensitivity" as requested.

  1. Spain, Greece, and France are still EU member states. I suggest the authors include here (as they have done in subsequent chapters) the EU provisions, or highlight that these individual member states have enacted specific provisions different from those of the EU, which is also not legally compliant. Thank you.

Response: Thank you for your insightful comment. To address the issue raised and clarify the legal status of MS-222 in the European Union (EU), we can revise the text to refer to EU provisions rather than specific member states, as you suggested. Here's the revised version:

MS-222 is approved as a fish anesthetic in several countries, including the United States, Norway, and New Zealand, with withdrawal periods of 21, 21, and 10 days, respectively, before consumption [38]. However, it is not approved in China or the EU, and no official EU standard methods for residue detection have been established. Despite this, MS-222 remains one of the most widely used anesthetics in global aquaculture, with established withdrawal periods and guidelines in regions where it is permitted [38]. The U.S. FDA allows its use in aquaculture with specific restrictions on fish families and requires adherence to withdrawal periods to ensure food safety.

  1. The authors refer to generic "aquatic species" and then include these two species, which the reader may not even be familiar with (are they fish? Crustaceans?). I suggest specifying what they are, thanks.

Response: Thank you for your comment. To improve clarity, we will specify the species mentioned, including their classification as fish, to ensure that readers can easily understand the context. Here is the revised version:

These properties make benzocaine an ideal anesthetic for a variety of aquatic species, including juvenile fish such as Colossoma macropomum and Aulonocara nyassae, providing a cost-effective alternative to other anesthetics [42,43].

  1. No dot before n 48, please.

Response: Thank you for your comment. We delete the dot before n 48, We also checked the entire text to avoid such mistakes.

  1. I checked the references and the publication reference is from 2018, therefore also from 2017. I ask the authors to verify whether anything has changed in these 7 years, in this regard, and to update their judgment. Thank you.

Response: Thank you. We conducted an updated literature and regulatory search to determine whether any significant changes have occurred in the past ~7 years (2018–2025) that would affect this statement. Our searches (PubMed, Web of Science, Scopus) did not reveal any convincing evidence that 2-phenoxyethanol has been newly approved for use in food‑fish aquaculture (i.e., species intended for human consumption) in major jurisdictions (e.g., EU, USA, Canada). Older studies note that 2‑PE “is not legally approved for use in aquaculture” for food fish because of safety/legality uncertainties (e.g., Misawa et al., 2014). While efficacy and dose‑response studies of 2-phenoxyethanol continue (e.g., a 2021 dose‑study: Rairat et al.) they focus mostly on efficacy, induction/recovery, or ornamental/experimental fish rather than food‑fish regulatory approval. We did not locate new regulatory bodies issuing MRLs (maximum residue limits) for 2-phenoxyethanol in edible fish, nor official guidelines updating its status to approved for food‑fish use. On the basis of the current evidence, we believe our original statement remains valid: 2‑phenoxyethanol is not approved for use in food‑fish aquaculture in the major jurisdictions we surveyed and remains generally recommended only for non‑food/ornamental contexts.

However, we admit that the research environment is constantly evolving, and a clear global regulatory investigation would be ideal. Therefore, we also stated in the last paragraph of 3.4:

“Currently, no countries explicitly authorize the use of 2-phenoxyethanol for fish intended for human consumption. Its use in food fish is prohibited in the U.S. and the European Union, where it is restricted to non-food fish, aquaculture research, and ornamental fish sedation. Concerns over food safety, legality, and environmental impact limit its broader application in aquaculture.”

If you believe we should focus more on specific regions or particular anesthetics as a potential area for future changes, we are open to further expanding on these aspects. Please feel free to share any additional suggestions.

  1. Please put the two fish species in italics.

Response: Thank you for your suggestion. We revised the two fish species in italics. We also checked the entire text to avoid such mistakes.

  1. Did the authors really mean to write eugenols in the plural or is it just a typo?

Response: Thank you for your comment. To avoid confusion, we correct it and list the related compounds as distinct entities.

Here is the revised version:

"Eugenol, methyleugenol, isoeugenol, methyl isoeugenol, eugenol acetate, and acetyl isoeugenol."

Reviewer 2 Report

Comments and Suggestions for Authors

See attached file

Comments on the Quality of English Language

A revision is ruquired.

Author Response

Reviewer 2: The information available on the use of anaesthetics in aquaculture fish remains scarce. Due to the wide variability observed among fish species (size, age, natural environment, performance, and physiology), the choice of type, dosage, stages of application, and duration of exposure to anaesthetics must be considered with utmost care. Thus, the topic addressed in this review is of the highest importance.

Response: Thank you for your insightful comment. We agree that the variability among fish species makes the use of anesthetics in aquaculture complex, and we have emphasized this aspect throughout the review. We believe this review provides important insights into the careful consideration required when choosing anesthetics, including dosage and exposure duration, to ensure their effective and safe application in different aquaculture species.

  1. The methodology selected by authors for the preparation of this review is not explained. Therefore, a section entitled methodology could be included. This section should state the key words used for the search, data bases screened, and the time frame chosen for the literature search. It should also explain the inclusion and exclusion criteria used for the selection of papers.

Response: Thank you. In response to your suggestion, we have added a section titled "Methodology" to provide clarity on the approach we used for this review. The methodology section now includes the key details regarding the literature search process, including the keywords, databases, time frame, and inclusion/exclusion criteria, ensuring transparency and reproducibility of the study selection process.

The updated methodology section reads as follows:

This review systematically identifies and analyzes relevant studies on fishery anesthetics, focusing on their safety, residue detection, and implications for food safety. A comprehensive literature search was conducted using keywords such as “fishery anesthetics,” “aquaculture anesthetics,” “residue detection,” and “maximum residue limits,” combined with Boolean operators, across databases including PubMed, Scopus, Web of Science, Google Scholar, and ScienceDirect. Studies published in peer-reviewed journals over the past three decades were included, as long as they addressed the use, regulation, safety profiles, and residue detection methods of anesthetics, incorporating both developed and developing countries. Non-peer-reviewed literature and irrelevant studies were excluded, and duplicate entries were removed during full-text screening. This methodology ensured the inclusion of relevant and up-to-date studies, providing a thorough analysis of commonly used anesthetics, recent advances in residue depletion, and detection techniques. Furthermore, the review identifies research gaps and suggests future directions to strengthen regulatory frameworks and safeguard public health. “On page 1-2 line 43-57.

  1. Line 5: This review examines six commonly 5 used anaesthetics in aquaculture—eugenol, MS

222, benzocaine, 2-phenoxyethanol, 6 diazepam, and quinaldine

Comment: Which was the criteria/rational to select these 5 anaesthetics? There are others, such

as metomidate and tricaine that are licensed for fish sedation and anaesthesia in salmon in Norway! Additionally, across Europe, the standard anaesthetic for fish is tricaine according to Schroeder et al. (2021). Tricaine is often referred as MS-222, this needs to be clarified in the manuscript from the beginning. (Schroeder, Paul, Richard Lloyd, Robin McKimm, Matthijs Metselaar, Jorge Navarro, Martin O’Farrell, Gareth D. Readman, Lars Speilberg, and Jean-Philippe Mocho. "Anaesthesia of laboratory, aquaculture and ornamental fish: Proceedings of the first LASA-FVS Symposium." Laboratory Animals 55, no. 4 (2021): 317-328.)

Response: Thank you for your insightful comment. The selection of the six anesthetics—eugenol, MS-222 (tricaine), benzocaine, 2-phenoxyethanol, diazepam, and quinaldine—was based on their representation in both scientific literature and common practices within aquaculture, as well as their relevance to fish welfare and food safety. While diazepam is indeed not commonly used in commercial aquaculture due to regulatory restrictions, it is frequently discussed in research settings and has been included in this review for its historical significance and its occasional use in experimental contexts.

The primary rationale for selecting these anesthetics was to cover a broad spectrum of substances commonly referenced in scientific studies and regulatory discussions. MS-222 (tricaine) is widely used in aquaculture and often considered the standard anesthetic, particularly in North America and Europe. Eugenol, benzocaine, 2-phenoxyethanol, and quinaldine are also well-established anesthetics with documented residue studies and regulatory considerations.

We acknowledge that other anesthetics, such as metomidate, are used in specific regions (e.g., Norway), but the focus on these six anesthetics allows us to provide a comprehensive review that reflects widely used substances and those for which residue depletion studies and detection techniques are most robust.

We have clarified in the manuscript that diazepam is discussed primarily in the context of its research use and historical applications, and that the selected anesthetics are representative of those most relevant to current aquaculture practices and regulatory frameworks. If you have any further suggestions or concerns, please feel free to raise them. We are committed to refining the manuscript and will make additional revisions as needed.

  1. Tricaine is often referred as MS-222, this needs to be clarified in the manuscript from the beginning.

Response: Thank you very much. We appreciate your suggestion to clarify the common name "tricaine" alongside the academic name "MS-222" from the outset. In response, we would like to explain our approach.

In the manuscript, we initially provide the full academic name, tricaine methanesulfonate (MS-222), on page 4, line 66, and later clarify that MS-222 is commonly referred to as tricaine in Section 3.2. Our intention is to first introduce the formal scientific name to maintain precision and then provide the common name in more detail when discussing the compound in depth, in order to avoid redundancy.

We believe this approach maintains the balance between academic precision and accessibility. If you have any further suggestions or concerns, we are happy to make additional revisions.

  1. Authors should carefully read the manuscript and add references whenever they refer to

other authors' findings or results.

Response: Thank you for your constructive feedback. We have carefully reviewed the manuscript and ensured that appropriate references are added whenever we refer to other authors' findings or results. If there are any specific sections where you feel references are still missing or need further clarification, please let us know, and we will be happy to make additional adjustments.

  1. Line 62: Figure 1 illustrates their chemical structures, aiding in understanding their molecular characteristics and potential residue behaviour.

Suggestion: Chemical structures could be discussed in more detail in order to clarify their potential impacts and to assist in the selection of anesthetics.

Response: Thank you for your suggestion. We have revised the manuscript to provide a more detailed discussion of the chemical structures of the anesthetics covered in the review. We have included discussions on how phenolic rings, ester groups, ether bonds, and lipophilic/hydrophilic properties in anesthetics like eugenol, MS-222, benzocaine, 2-phenoxyethanol, and quinaldine contribute to their efficacy, toxicity, and residue elimination. The modified contents are as follows:

Eugenol (C₁₀H₁₂O₂) is a naturally occurring phenolic compound predominantly found in various essential oils, especially clove oil, where it constitutes 85-95% of the active ingredient, along with minor components such as methyl eugenol and isoeugenol [9,10]. Its chemical structure consists of a phenolic ring and an ethylene group, contributing to its water solubility. Eugenol functions as an effective anesthetic by inducing temporary immobility in farmed fish, thereby reducing stress responses during procedures such as transportation, handling, and artificial reproduction. This results in a reduction of physical injuries, including mechanical damage and scale loss, and minimizes the risk of subsequent infections [7]. “on page 5, line 81-89.

MS-222 (3-aminobenzoic acid ethyl ester methanesulfonate), commonly referred to as tricaine, possesses a chemical structure comprising a benzene ring and an amino acid ester group. It is absorbed into the bloodstream through the gills and skin and is subsequently distributed throughout the body [32]. MS-222 induces anesthesia primarily by blocking sodium ion channels in muscle cells and, to a lesser extent, potassium ion channels in nerve membranes. This mechanism results in a loss of consciousness and a reduction in metabolic activity [33]. “on page 6, line 129-135.

Benzocaine (C₉H₁₁NO₂) is a lipophilic compound widely used as a local anesthetic, renowned for its rapid onset, low toxicity, stable efficacy, and prolonged effects. Its chemical structure consists of a benzene ring and an ester group, which enable it to penetrate cell membranes and interact with the nervous system. Benzocaine exerts its anesthetic effect by binding to sodium ion channels, thereby blocking nerve impulse transmission. These properties make benzocaine an ideal anesthetic for a variety of aquatic species, including juvenile fish such as Colossoma macropomum and Aulonocara nyassae, providing a cost-effective alternative to other anesthetics [42,43]. “on page 8, line 174-181.

2-Phenoxyethanol is a colorless, oily liquid that is soluble in water and commonly used as a local anesthetic in aquaculture. Its chemical structure consists of a phenoxy group (C8H10O2) attached to an ethanol group, forming an ether bond between the oxygen atom of the phenoxy group and the ethyl chain of the ethanol group [50]. This molecular configuration gives 2-phenoxyethanol both lipophilic and hydrophilic properties, enabling it to pass through biological membranes efficiently. In fish, it is absorbed through the gills and skin, subsequently transported via arterial blood to the central nervous system, with excretion predominantly occurring through branchial respiration. In rainbow trout, its biological half-life is approximately 30 minutes [51]. “on page 9, line 205-213.

Diazepam is a long-acting benzodiazepine sedative that acts as a positive allosteric modulator of type-A γ-aminobutyric acid receptors (GABAARs), enhancing the effect of γ-aminobutyric acid (GABA) and reducing neuronal activity [62]. It is commonly used to treat neurological disorders such as epilepsy, anxiety, and sleep disturbances. “on page 11, line 245-247.

Quinaldine (2-methylquinoline) is an alkaloid anesthetic with antibacterial and antipyretic properties, widely used as a precursor for the synthesis of various pharmaceuticals, including biocides and bactericides [70]. Its chemical structure consists of a quinoline ring system, a bicyclic structure comprising a benzene ring fused to a pyridine ring, with a methyl group (-CH₃) attached to the second position of the quinoline nucleus. The presence of a nitrogen atom in the pyridine ring contributes to its basicity, allowing quinaldine to interact with biological systems through ion exchange and receptor binding. This molecular configuration imparts lipophilicity to quinaldine, enabling efficient absorption through cellular membranes. Since the 1950s, quinaldine has been investigated as a fish anesthetic and has since been widely applied in fisheries for processes such as capture, transportation, sampling, and measurement [70]. “on page 12, line 292-299.

  1. Line 81: Studies on eugenol in Pacific white shrimp revealed that, regardless of treatment

duration (e.g., 300 mg L-1 for 5 minutes or 10 mg L-1 for 24 hours), residual eugenol concentrations fell below 2.5 mg kg-1 within 24.5 hours. Comment: Please add a reference. Line 84: within 24.5 hours. Comment: Please add a reference.

Response: Thank you for your comment. We have added the appropriate references to support the information provided in the manuscript.

  1. Line 50: 2. Overview of Fishery Anesthetics

Suggestion: Anesthetics are used on farmed fish. So maybe it's better to use farmed fish instead of fish in the whole manuscript.

Response: Thank you for your suggestion. We agree that the term "farmed fish" is more accurate and specific in the context of aquaculture. In response, we have revised the manuscript to replace "fish" with "farmed fish" in the text to better reflect the focus on species used in aquaculture practices. For example:

Fishery anesthetics are extensively employed in aquaculture to mitigate stress and reduce mortality during handling, transportation, and processing of farmed fish. In line 8-9.

Fishery anesthetics are chemical agents employed in aquaculture to induce sedation or temporary immobility in farmed fish, thereby reducing stress and physical injury during operations such as transport, handling, sampling, and artificial reproduction [6,7]. On page 2 line 59-61.

  1. Line 69: 3. Pharmacological Actions and Safety Assessment. This means Safety Assessment of Fish? Or Safety Assessment of Fish as Food?

It is important to clarify if the aim is to discuss the safety of animals to which anaesthetics are administered? Or the potential impact in terms of food safety?

Response: Thank you for your valuable feedback. We understand the need for clarification regarding the focus of the safety assessment in section 3. We have revised the wording in section 3 to clearly indicate that the safety assessment refers to both the safety of the fish (as animals) and the potential impact on food safety.

In the revised manuscript, we have clarified that the section addresses:

  1. The safety of the fish receiving anesthetics (i.e., the potential physiological and toxicological effects on the animals).
  2. The food safety implications, particularly in relation to residual anesthetics and their metabolites in edible tissues, which may affect human consumers.

Here is the revised sentence:

Additionally, the half-life of eugenol in seabass varies with water temperature, with faster metabolism observed at higher temperatures (e.g., 2 hours at 20°C vs. over 4 hours at 13°C) [16]. These findings suggest that controlling water temperature can accelerate eugenol metabolism, enhancing residue elimination from fish tissues and supporting food safety.

Although eugenol is effective as an anesthetic, it poses cytotoxic risks to fish at high concentrations. In vitro studies have shown that eugenol can cause developmental abnormalities in zebrafish embryos, including skeletal deformities and pericardial edema, when exposed to concentrations between 0.5 and 2 mg L⁻¹ [17]. Tao et al. also found that eugenol exposure delayed zebrafish embryo hatching, reduced body length, and decreased the inflation rate of swim bladders [18]. Moreover, eugenol has been linked to cytotoxicity in various cell lines [19], including mouse fibroblasts [20], rat hepatocytes [21], and human dental pulp fibroblasts [22,23]. In vivo studies also suggest its potential for oral mucosal damage in rats [24,25]. However, despite these cytotoxic risks, studies on rats and other animals have shown no significant toxicity at low concentrations, further supporting the safety of eugenol when used according to recommended guidelines [26]. On page 5 line 99-114.

While MS-222 is generally safe for use in fish, prolonged exposure in humans, particularly through the consumption of fish with high residual concentrations, can cause skin and respiratory irritation, and in rare cases, retinal damage [40]. However, its high water solubility and rapid metabolism in fish ensure that it typically does not accumulate in edible tissues. When proper withdrawal periods are observed, the residue levels of MS-222 remain sufficiently low to pose minimal risk to human health. Studies have shown that MS-222 significantly reduces stress in fish during anesthesia, thereby enhancing animal welfare in aquaculture operations [41]. Although MS-222 has been shown to be toxic to certain fish species at high concentrations, its high water solubility and the absence of documented human health risks further support its use as a safe anesthetic when the recommended withdrawal period is followed. To ensure food safety, it is crucial to monitor exposure levels, particularly in species that may have heightened sensitivity to MS-222. On page 7 line 150-161.

Despite its generally low toxicity to fish, excessive human exposure, such as through the consumption of fish with high residual concentrations, can lead to toxicity, including arrhythmias, coma, and pulmonary complications [46]. It may also cause allergic reactions, such as contact dermatitis and hypersensitivity [47]. However, benzocaine's rapid metabolism and excretion through the gills typically result in low residue levels in fish tissues. When proper withdrawal periods are observed, these residues remain sufficiently low to pose minimal risk to human health. Genotoxicity studies have demonstrated that benzocaine does not possess genotoxic properties, further supporting its safety for human consumption when used according to established guidelines [48]. On page 8 line 190-198.

  1. Line 292: anesthetic residues are usually present at low concentrations, and their metabolites may exhibit toxic or pharmacological effects, necessitating highly sensitive and reliable detection methods. The authors intend to consider which residues? Please precise!

Response: Thank you. To address your query, we have clarified which specific anesthetic residues are considered in the manuscript. Here is the revised sentence:

Moreover, anesthetic residues, such as those from eugenol, MS-222, benzocaine, 2-phenoxyethanol, Diazepam and their metabolites, are usually present at low concentrations. These residues and metabolites may exhibit toxic or pharmacological effects, necessitating highly sensitive and reliable detection methods. On page 15 line 379-383.

  1. Among the pre-treatment and extraction methods evaluated by the authors, which were considered most effective for the different types of residues?

Response: Thank you. we have revised the manuscript to provide a clearer summary of the most effective methods for detecting different anesthetic residues.

We have specified that GC-MS and LC-MS/MS are the most widely used and stable techniques for anesthetic residue detection, particularly due to their ability to accurately analyze both anesthetics and their metabolites. We also mentioned that the selection of pre-treatment methods depends on the anesthetic type, as each anesthetic exhibits unique chemical properties that require tailored extraction techniques. For example, GC-MS is highly effective for detecting volatile anesthetics like eugenol and MS-222, while LC-MS/MS is more suitable for non-volatile anesthetics like 2-phenoxyethanol and benzocaine.

Furthermore, we highlighted that effective pre-treatment extraction methods, such as SPE (solid-phase extraction) and SPME (solid-phase microextraction), are crucial for minimizing interference and enhancing the sensitivity of residue analysis. We also acknowledged that the development of optimal pre-treatment methods and materials remains an important area of research to ensure the reliability and accuracy of anesthetic residue detection. Here is the revised sentence:

In summary, to effectively detect anesthetic residues, it is essential to match the anesthetic type with the appropriate pre-treatment method. GC and GC-MS are among the most widely used methods for anesthetic residue analysis due to their high sensitivity, effective separation capabilities, and precise quantification of trace anesthetics. These methods are especially effective for detecting volatile or semi-volatile anesthetics like eugenol and MS-222. LC and HPLC are particularly suitable for anesthetics like 2-phenoxyethanol and benzocaine, which are commonly used in aquaculture but are more challenging to analyze using GC. GC-MS and LC-MS/MS remain the most stable and sensitive techniques for residue detection, partly due to their ability to stably analyze anesthetic and anesthetic metabolites, which are crucial for understanding the full scope of anesthetic exposure and residue presence. However, the development of effective pre-treatment extraction methods and the use of appropriate pre-treatment materials continue to be important areas of research to ensure the reliability and accuracy of these techniques. On page 19 line 465-477.

  1. Thus, it was expected that the authors would discuss (i) the suitability of the most commonly

used anesthetics for different aquatic farmed species, taking into account environmental the

main objective to apply these substances, environmental conditions (e.g. water quality, including

temperature and pH), use of good practices and potential impacts on food safety, (ii) suggest (if

possible) the most suitable ones for selected species (taking into account the specific production

conditions),

Response: Thank you. We have added a new section, Section 4: Considerations on the applicability of commonly used anesthetics in aquaculture, to address your suggestion. In this section, we provide a comprehensive analysis of the suitability of anesthetics for different farmed aquatic species, with particular emphasis on environmental factors (such as water quality, temperature, and pH), and the associated impacts on the application and food safety.

Furthermore, we discussed the potential impacts on food safety, highlighting the need for appropriate withdrawal periods and residue monitoring to minimize risks. The section also suggests the most suitable anesthetics for different species based on specific environmental conditions (e.g., eugenol for tropical fish species like tilapia and guppy, and MS-222 for cold-water species like rainbow trout and salmon).

Thank you again for your insightful comment, and we believe this addition significantly improves the clarity and depth of our review. We are happy to make further adjustments if needed.

  1. It would also be important for authors to mention the latest advances and the most urgent needs of aquaculture producers. Finally, it would also be expected that the manuscript would indicate some future research needs, based on the outputs of this review, to find robust solutions from a scientific and practical point of view to overcome the current difficulties identified. Thus, the manuscript presents some aspects that require further consideration. Therefore, it is suggested that the authors conduct a thorough review and resubmit the manuscript for evaluation.

Response: Thank you. In response, we have revised the manuscript to highlight key issues, including the development of new anesthetic formulations, urgent needs for aquaculture producers, and future research directions. Here is the revised sentence:

Fishery anesthetics, such as eugenol, MS-222, and benzocaine, are essential tools in aquaculture, providing effective methods for reducing stress and improving animal welfare during farming operations. However, significant challenges remain due to the lack of harmonized regulations across regions and concerns about the accumulation of residues in edible tissues, which can impact both food safety and consumer confidence. The illegal use of certain sedatives, such as diazepam, further complicates the situation, raising concerns about bioaccumulation and long-term food safety risks.

Looking ahead, several areas of research are critical to advancing the field. First, the development of new, eco-friendly anesthetic formulations that are both low in toxicity and capable of rapid biodegradation is a key priority. Such formulations would reduce the risk of residual buildup in aquatic organisms, addressing both environmental and food safety concerns. Second, advancements in multi-residue detection technologies, such as the development of more sensitive biosensors and point-of-care testing methods, will enable faster, more efficient monitoring of anesthetic residues in aquatic products.

Moreover, the adoption of a One Health approach—integrating human, animal, and environmental health considerations—is essential for ensuring comprehensive safety in aquaculture. Globally harmonized regulatory frameworks will play a pivotal role in reducing trade barriers, facilitating safer and more sustainable aquaculture practices across different regions. Equally important is the promotion of responsible aquaculture practices through targeted training programs for producers, which will help minimize anesthetic use and encourage more sustainable farming methods. On page 23 line 588-608.

Round 2

Reviewer 2 Report

Comments and Suggestions for Authors

The suggestions have been incorporated and the changes made are appropriate. Overall, the manuscript has been significantly improved, so as far as I am concerned, I consider the revision complete.